# The Effect of Academic Environment on Turnover Intention of High-Skilled Scientific and Technology Professionals: Empirical Evidence from China’s First-Class Universities

**DOI:** 10.3390/bs15111475

**Published:** 2025-10-29

**Authors:** Xiuliang Dai, Lijian Wang, Dan Chen, Xiaoyu Guo

**Affiliations:** School of Public Policy and Administration, Xi’an Jiaotong University, Xi’an 710049, China; wanglijian2@mail.xjtu.edu.cn (L.W.); cd1429149424@stu.xjtu.edu.cn (D.C.); guoxiaoyu@stu.xjtu.edu.cn (X.G.)

**Keywords:** high-skilled scientific and technology professional, academic environment, job satisfaction, turnover intention

## Abstract

The study constructs a theoretical framework for how the academic environment and job satisfaction affect the turnover intention of high-skilled science and technology talents (STPs), based on academic ecology theory and job satisfaction theory. Conducting a quantitative analysis based on survey data collected from 482 national-level STP recipients across 14 first-class universities, the study integrates descriptive statistics, ANOVA, OLS regression, and bias-corrected bootstrap mediation analysis to examine the relationships between these variables. Findings reveal that STPs exhibit relatively low overall turnover intention; however, significant regional disparities exist, with higher turnover intentions observed in central and western regions. Academic environment shows moderate overall positivity, with competition receiving the highest mean score. Both academic environment and job satisfaction significantly and negatively predict turnover intention. Furthermore, job satisfaction fully mediates the relationship between collaboration/democracy and turnover intention. Based on the findings, the study recommends that universities strengthen the institutional development of the academic environment to provide a favorable environment for talent development and scientific and technological innovation.

## 1. Introduction

Science and technology professionals (STPs) are the driving force behind national innovation and development, primarily referring to individuals engaged in specialized academic research in universities and various research institutions. Across the globe, nations consider the nurturing and elevation of high-skilled professionals to be a pivotal strategy for enhancing their overall national competitiveness ([31]). Since the 1990s, China has embarked on a bold endeavor, launching a suite of innovative national-level programs and initiatives, notably the esteemed Chang Jiang Scholars Program, to cultivate and bolster a distinguished cadre of talent. The country has also established a comprehensive financial support system that caters to scientific professionals of all ages, encompassing both those recruited internationally and those developed domestically ([13]). These STPs, selected for the National Talent Program, represent the frontier level of China’s scientific and technological innovation in specific fields and are an important force to promote China’s scientific and technological progress. At the same time, under the background of China’s efforts to build world-class universities, the demand for high-skilled professionals in universities is increasing ([18]), leading to more frequent job mobility of high-skilled STPs between universities. An insightful study reveals intriguing trends in this field. Among the elite group of 3796 recipients of the prestigious National Science Foundation for Distinguished Young People prior to and including 2017, a notable 567 had transitioned to new employment institutions by May 2018. And this migration pattern highlights a clear trend: these talented professionals are gravitating towards economically vibrant regions in the Southeast, attracted by strong research environments, advanced infrastructure, and promising career opportunities ([9]).

Functional talent mobility often emerges organically from professionals’ career development trajectories. Such structured mobility patterns further contribute to optimizing human capital allocation ([39]). However, China’s evolving market economy and intensified inter-university competition for professionals have complicated talent mobility patterns. This trend is particularly pronounced in the central and western regions, where higher education institutions experience disproportionate outflows of STPs, resulting in irregular migration dynamics ([38]). From a macro perspective, regional socioeconomic disparities, institutional dysfunctions, and inadequate management practices are the primary factors driving the flow of STPs ([39]). From a micro perspective, altering jobs or workplaces represents a rational decision made by individuals, influenced by a myriad of factors. Existing research has predominantly concentrated on the influence of job satisfaction on individual’s turnover intention ([1]; [3]).

Unlike conventional professions, highly skilled STPs possess not only deep domain expertise but also demonstrate strong innovative capacities. Their primary research objective centers on achieving breakthrough innovations. From an innovation ecosystem theory perspective, a robust academic ecosystem constitutes a critical foundation for technological advancement and talent development ([32]). On one hand, it furnishes a conducive environment and robust institutional framework for research endeavors. On the other hand, this strategy enhances the perceived value of research activities among individual scholars ([12]). Cultivating a robust academic ecosystem has emerged as a critical strategic objective within the governance frameworks of modern universities ([25]; [36]). An academic ecosystem is fundamentally structured around three interrelated components: research actors, knowledge outputs, and the academic environment. Research actors refer to individuals or collectives actively involved in knowledge production and innovation. Knowledge outputs constitute the tangible manifestations of scholarly innovation, such as peer-reviewed publications, patents, and monographs. The academic environment encompasses the intellectual and operational context in which researchers operate, influenced by external policy frameworks and institutional governance systems. Furthermore, this environment exerts a substantial influence on both the innovation trajectories and productivity outcomes of research actors

According to the established scholarly literature, the academic environment encompasses critical dimensions, including integrity norms ([5]), democratic discourse ([28]), equitable competition frameworks, and structured collaboration mechanisms ([11]; [33]). In 2007, the Chinese Academy of Sciences issued the pioneering Opinions on Strengthening the Construction of Scientific Research Integrity, emphasizing the imperative to cultivate an academic ecosystem balancing competitive dynamism with collaborative symbiosis. The document articulates that researchers should engage in complementary and interdependent collaborations to achieve collective advancement. Furthermore, it mandates establishing an academic culture characterized by structured competition and comprehensive cooperation, empowering elite technological talent to drive scholarly innovation.

However, scholarly attention to the academic environment within Chinese universities in the contemporary context remains limited, particularly regarding its impact on job satisfaction and turnover intention. This study examines how the academic environment at China’s first-tier universities influences turnover intention among highly skilled STPs. The specific research questions addressed are as follows: (1) What constitutes the current state of the academic environment in China’s Double First-class universities? (2) To what extent does the institutional academic environment affect turnover intention of high-skilled STPs? (3) Does job satisfaction mediate the relationship between academic environment and turnover intention? This study addresses the lack of attention to the relationship between the academic environment in universities and turnover intention of STPs and contributes empirical evidence to university talent management and academic innovation.

## 2. Literature Review and Research Hypothesis

### 2.1. The Relationship Between Academic Environment and the Turnover Intention of High-Skilled STP

The term “environment” fundamentally refers to the living conditions and inter dependencies among organisms ([14]). From the perspective of the academic ecology theory, the academic environment directly influences the production of innovative outcomes. A supportive academic environment can stimulate researchers’ creativity and foster collaboration among individuals. However, this theory has not yet addressed the effect of the micro-level academic environment on individual career development ([37]). The academic environment can be conceptualized as a dynamic ecosystem shaped by the interactions and mutual influences among researchers and their activities within the complex milieu of academic inquiry ([15]). Although influenced by the scientific research management system, the academic environment remains distinct from both institutional governance structures and formal academic norms. Prior research has documented persistent scientific misconduct, which inevitably undermines the integrity and reputation of academia. In recent years, an alarming rise in such misconduct—including high-profile academic scandals—has been observed globally, with China being no exception ([37]).

STPs are a crucial factor in scientific and technological innovation. Countries around the world generally regard the cultivation and attraction of high-skilled STPs as a key strategy for enhancing their competitiveness in science and technology. Therefore, the professional mobility of STPs also constitutes an essential pointin the development of the academic ecosystem. The academic environment encompasses researchers’ perceptions of their domain and the intangible scholarly atmosphere in which they operate. Unlike general material production, scientific research and innovation result from the interaction between researchers’ active creativity and their social environment. This process necessitates a conducive innovation environment characterized by competition, collaboration, democracy, and integrity ([7]). When STPs perceive their academic environment as detrimental to the development and growth of their scientific research and innovation activities, they may seek a more favorable environment through turnover. Thus, Hypothesis 1 is proposed.

**H1:** 
*The more positively high-skilled STPs view the academic environment, the less inclined they are to contemplate turnover.*


### 2.2. The Relationship Between Job Satisfaction and the Turnover Intention of High-Skilled STP

Talent mobility within defined geographic and sectoral boundaries is driven by labor market supply-demand dynamics and individual career development choices. This movement fundamentally represents the reallocation and circulation of knowledge resources across regions and industries. For highly skilled technology professionals, strategic career transitions constitute an essential developmental stage ([35]).

Job satisfaction theory is a significant area of study in organizational behavior and human resource management, aiming to explain employees’ subjective perceptions and evaluations of their work and work environment ([22]). Job satisfaction significantly influences employees’ attitudes and behaviors, primarily reflected in aspects such as work performance, work engagement, and turnover intentions ([1]; [16]). From a rational choice perspective, individuals’ job satisfaction increases when their organizational and work-related needs are fulfilled. This satisfaction typically manifests through enhanced work engagement, prompting reciprocal contributions to the organization. Conversely, diminished job satisfaction often correlates with turnover intentions. The inverse relationship between job satisfaction and turnover propensity is well-established in empirical research ([16]; [22]; [34]).

Job satisfaction remains a pivotal construct in organizational behavior research investigating talent mobility. This theoretical framework has proven valuable in analyzing faculty turnover dynamics within higher education institutions ([29]). Therefore, examining the impact of job satisfaction on turnover intention among highly skilled technology professionals is critical. Prior studies have treated job satisfaction as an undifferentiated construct, whereas this research adopts a multidimensional approach. We assess five specific dimensions affecting work–life integration: compensation, living conditions, workplace environment, institutional reputation, and administrative support. This granular analysis verifies how job satisfaction influences mobility decisions. Thus, Hypothesis 2 is proposed.

**H2:** 
*The greater the job satisfaction among high-level technology talents, the less likely they are to consider turnover.*


### 2.3. The Mediating Role of Job Satisfaction

The academic environment provides an external context conducive to research activities among high-skilled STPs and plays a critical role in their engagement with scientific innovation. Job satisfaction reflects the alignment of compensation, living conditions, workplace atmosphere, institutional reputation, and administrative support with STP’s personal requirements. According to the Job Demands-Resources (JD-R) Model ([6]; [8]), supportive aspects of the academic environment can mitigate work-related stress, enhance job satisfaction, and thereby reduce turnover intentions. Conversely, the absence of adequate support in academic settings may amplify intrinsic stress and drive employees toward turnover decisions. When high-skilled STP experience maladaptation to their academic environment, this is often manifested through dissatisfaction with external conditions such as workplace environment, salary, and management systems. Existing studies have also confirmed that a positive academic environment can enhance researchers’ job satisfaction ([23]; [24]). Based on this theoretical foundation, we propose the Hypothesis 3:

**H3:** 
*Job satisfaction plays a mediating role between academic environment and turnover intention of high-skilled STPs.*


Based on the aforementioned analysis, the relationship between the academic environment, job satisfaction, and the turnover intention of high-skilled STPs is shown in Figure 1.

## 3. Data, Variables, and Method

### 3.1. Data Source

The high-skilled STPs studied in this study are those admitted into national-level talent-supporting programs and projects, including the Distinguished Professorship Project, the Chair Professorship Project, the Young Scholars Project of the Chang Jiang Scholars Program, the Introduction Program of High-caliber Overseas Talents (also known as the “Thousands of Talents Program”), the Outstanding Talents Project, the Leading Talents Project, and the Young Top-tier Talents Project of the Ten Thousands of Talents Program, the Excellent Young Scientists Fund, and the National Science Fund for Distinguished Young Scholars.

From May to July 2020, under the funding support of the National Natural Science Foundation of China (NSFC) project titled “Research on National Funding System for Young Scientific and Technological Talents,” the research group selected 14 Double First-class universities to conduct questionnaire surveys. These institutions were distributed across China, with seven located in Eastern China, four in Central China, two in Western China, and one in Northeast China.

A combination of quota sampling and random sampling was applied in the survey. Respondents comprised both STPs admitted into national-level talent-supporting programs and those who had not been admitted. The survey covered fundamental characteristics of STPs, participation in national-level talent-supporting programs, motivation for achievements, job satisfaction, turnover intention, and evaluation of the academic environment at their affiliated universities. With support from the National Natural Science Foundation of China, the research group distributed questionnaires to STPs at these universities via email. To ensure the reliability of the survey data, we informed participants of the purpose and content of the survey, and their participation was voluntary. We also reviewed the collected questionnaires and excluded samples with missing critical information. Ultimately, a total of 897 valid questionnaires were submitted, including 482 responses from national-level sci-tech talents. The geographic distribution of respondents across China was as follows: 281 were affiliated with universities in eastern China, 122 in central China, and 79 in western China. The study utilized SPSS 22, along with its PROCESS macro for data analysis.

### 3.2. Variable Measurement

(1) Turnover intention. In some studies, job mobility and turnover intention are used interchangeably. This study proposes that turnover intention be defined specifically as an individual’s intention to transition between academic institutions, entailing a change in their university employer while maintaining the same job function. STP competition has predominantly occurred among universities, especially under the Double First-class University Initiative. Consequently, greater attention merits the inter-university mobility of high-caliber tech talent.

This study operationalizes the turnover intention of high-skilled STPs through a structured survey item measuring their “possibility of switching university employers.” Responses were captured on a five-point Likert scale: Very Unlikely (1), Unlikely (2), Neutral (3), Likely (4), and Very Likely (5).

(2) Academic environment. As of now, no quantitative studies have specifically examined academic environment within university settings. As defined earlier in this study, academic environment refers to the perception of the academic environment held by STP employed by the university. Consequently, the university’s academic environment is evaluated based on these researchers’ perceptions of its academic atmosphere, encompassing dimensions of competition, collaboration, democracy, and integrity. This evaluation is operationalized through responses to the following question: “How would you rate the atmosphere of competition, collaboration, democracy, and integrity at your university?” Responses are recorded on a five-point Likert scale: Very Poor (1), Poor (2), Average (3), Good (4), and Very Good (5).

(3) Job Satisfaction. Job satisfaction, frequently conceptualized as a broad construct in prior research, encompasses multifaceted personal needs, particularly among high-skilled STPs. Consequently, understanding their job satisfaction necessitates a multi-perspective approach, specifically examining their scientific work and research context. This study operationalizes job satisfaction through five distinct dimensions: Satisfaction with salary, Satisfaction with living environment, Satisfaction with work environment, Satisfaction with the university’s reputation, and Satisfaction with management services. Responses are rated on a five-point Likert scale: Very Dissatisfied (1), Dissatisfied (2), Neither Dissatisfied nor Satisfied (3), Satisfied (4), Very Satisfied (5).

(4) Controlled variables. Building upon established practices in variable control within prior research ([1]; [17]), this study operationalizes individual factors of STPs as control variables, including gender, years of working experience, geographical location of the university, and the establishment level of research platforms.

### 3.3. Research Method

(1) Descriptive statistics. This study employs descriptive statistics to analyze the levels of turnover intention, perceived academic environment, and job satisfaction among high-skilled STPs. (2) Analysis of Variance. Analysis of variance (ANOVA) is applied to examine potential regional differences in these variables. (3) Multiple Linear Regression. To investigate the direct effects of the academic environment and job satisfaction on STP turnover intention, OLS analysis is utilized. (4) Mediation Analysis. Furthermore, the mediating role of job satisfaction in the relationship between the academic environment and turnover intention is tested using the bias-corrected Bootstrap method ([27]). This mediation analysis technique, introduced by Preacher and Hayes, offers advantages over traditional regression-based approaches and the Sobel test by directly assessing the significance of the mediation pathways.

To address severe multicollinearity among the four dimensions of academic environment in the regression model, dimensionality reduction was performed using factor analysis. The data met the criteria for factor analysis. The extracted principal component accounted for 79.56% of the total variance, with each dimension contributing over 76% to this component. Consequently, the averaged values of the four dimensions were incorporated into the regression analysis.

## 4. Result Analysis

### 4.1. Turnover Intention, Perceived Academic Environment, and Job Satisfaction Among High-Skilled STPs

Table 1 presents descriptive statistics for turnover intention, perceived academic environment, and job satisfaction among high-skilled STPs. The mean turnover intention score is 2.35, indicating a relatively low level. Significant regional differences in turnover intention were observed (*p* < 0.05), with STPs in eastern universities demonstrating the lowest likelihood of turnover compared to other regions.

STPs perceive their academic environment as moderately positive. As shown in Table 1, the mean scores for perceived collaboration, competition, democracy, and integrity dimensions are 3.33, 3.94, 3.23, and 3.29, respectively. No significant regional differences (*p* > 0.05) were observed in the STP’s perception of academic environment. However, comparative analysis across dimensions revealed that the competition atmosphere (mean = 3.94) was significantly higher than all other dimensions.

Overall, high-skilled STPs express relatively high job satisfaction, with an average salary satisfaction score of 3.21. Notably, eastern region talents report significantly lower salary satisfaction than those in the central and western regions, likely due to higher living costs and work pressure despite higher salaries. Conversely, eastern talents exhibit greater satisfaction with work environments (overall 3.42) and management services (overall 3.31), alongside the highest university reputation scores (3.51), attributed to the region’s advanced higher education infrastructure. While living environment satisfaction remains balanced nationally (3.39), cross-dimensional analysis reveals higher satisfaction with university reputation and living conditions compared to comparatively lower scores for salary and management service satisfaction.

### 4.2. The Effect of Academic Environment and Job Satisfaction on the Turnover Intention of High-Skilled STPs

Table 2 presents the regression analysis results examining the effects of academic environment and job satisfaction on the turnover intention of STPs. In Model 1, the independent variable academic environment demonstrates a significant negative effect on STP turnover intention. The regression results indicate that, holding other variables constant, a one-unit increase in the academic environment score is associated with a 0.146-unit decrease in turnover intention. This finding suggests that a favorable academic environment contributes to reducing STP’s inclination to leave. Among the control variables, university geographical location also significantly influences turnover intention. Statistical results reveal that STPs working in central and western regions exhibit a higher propensity for mobility compared to their counterparts in the eastern region. This regional disparity aligns with the findings from the descriptive statistics and reflects the prevailing trend of inter-regional talent flow within China.

Model 2 examines the effect of job satisfaction on the turnover intention of STPs. Regression results indicate that satisfaction with salary, work environment, and university reputation significantly influences turnover intention. Holding other variables constant, higher satisfaction levels correlate with a lower likelihood of STP mobility to other institutions. Furthermore, a comparison of regression coefficients across job satisfaction dimensions reveals that work environment satisfaction exerts the strongest effect on individual turnover intention, followed by salary level. The influence of geographic location remains statistically significant.

Model 3 simultaneously incorporates both academic environment and job satisfaction in the regression analysis. The results indicate that while the effect of job satisfaction on turnover intention remains statistically significant, the effect of academic ecology is not significant. This pattern suggests that job satisfaction may mediate the relationship between academic ecology and flow intention.

### 4.3. Mediating Effect Test of Job Satisfaction

Table 3 presents the results of the mediating effect test for job satisfaction, demonstrating both the direct effects of collaboration, competition, democracy, and integrity on the turnover intention of high-skilled STPs, and the mediating role of their job satisfaction. In this analysis, gender, years of working, university location, and research platform level were included as control variables. According to the bias-corrected bootstrap method for testing mediation effects, a direct or indirect effect is not statistically significant if its confidence interval includes zero. When both direct and indirect effects are significant, partial mediation occurs. In contrast, if the direct effect is non-significant while the indirect effect remains significant, complete mediation is established.

The mediation analysis reveals that job satisfaction fully mediates the relationship between academic collaboration and turnover intention, as well as between academic democracy and turnover intention. In contrast, it partially mediates the effects of academic competition and academic integrity on turnover intention. These findings suggest that a more positive perception of the academic environment, combined with higher levels of job satisfaction, significantly reduces the likelihood of talent mobility to other institutions.

## 5. Discussion and Conclusions

High-skilled STPs serve as the driving force for innovation-driven development in science and technology and constitute the core component for universities to enhance their global competitiveness ([30]; [32]). A sound academic environment is a prerequisite for scientific and technological innovation and for the cultivation and development of talent ([3]; [4]). Based on the data from a sample survey of high-skilled STPs at universities under the Double First-Class Initiative in China, this study examines current patterns of turnover intention, job satisfaction of high-skilled STPs, and academic environment in Double First-Class universities, while statistically verifying (a) the direct effects of academic environment and job satisfaction on turnover intention and (b) the mediating role of job satisfaction in the academic environment–turnover intention relationship.

This study reveals that the overall turnover intention of high-skilled STPs is relatively weak, as they generally benefit from competitive salaries and institutional support, making them a key target for talent cultivation programs ([21]). However, regional disparities in economic development and the implementation of the Double First-Class Initiative have led to divergent mobility patterns: talents working in universities located in western and central China exhibit significantly stronger turnover intentions compared to those in eastern regions, due to limited research infrastructure, fewer career advancement opportunities, and weaker institutional competitiveness ([19]; [39]).

An interesting finding is that STPs in different regions exhibit varying levels of job satisfaction. Specifically, those in the eastern region, report the lowest satisfaction with salary, those in the central region show the lowest satisfaction with management services, and those in the western region express the lowest satisfaction with the work environment. This suggests that the impact of regional socioeconomic development levels on individual job satisfaction is significantly different. The relatively high cost of living in the eastern region makes individuals place greater emphasis on salary levels, while the central and western regions still need to improve their management services and infrastructure development.

In recent years, there have been many discussions about academic environment in the academic circle ([36]; [37]) but few about the relationship between academic environment and talents’ turnover. This study operationalizes academic environment as the talent’s perception of the academic atmosphere in their universities, which includes collaboration, competition, democracy and integrity. We found that academic environment significantly affects the turnover intention of high-skilled STPs. This important finding has been overlooked in previous research. Adequate funding support and advanced hardware facilities form the foundation for conducting innovative research ([20]), while a healthy academic environment is equally vital. On one hand, a democratic and integrity-driven academic environment can stimulate researchers’ motivation for academic achievement, fostering self-identity in scientific endeavors and contributing to the development of the academic community ([10]). On the other hand, structured cooperation and competition are essential for academic network growth—such balanced dynamics not only enhance individual creativity but also promote the holistic development of research teams ([26]). Additionally, the study identifies a mediating mechanism through which the academic environment influences turnover intention. Specifically, the higher high-skilled STPs’ perception of the academic environment, the greater their job satisfaction, ultimately leading to reduced turnover intention.

Job satisfaction plays a crucial role in shaping turnover intention, consistent with prior studies ([2]; [16]; [34]). This study advances existing studies by decomposing job satisfaction into multiple dimensions—specifically salary, work environment, living environment, university reputation, and management service—and identifying these as critical factors influencing turnover decisions. A key finding is that high-skilled STPs in Chinese universities across different regions exhibit significant variations in job satisfaction. Specifically, the eastern region sample demonstrates higher job satisfaction, which may be attributed to the imbalance in regional socioeconomic development.

The findings of this study provide significant implications for optimizing academic environment construction and talent cultivation systems in universities. First, it is essential to improve the academic governance system in higher education institutions and emphasize the critical role of talent in scientific and technological innovation. Within key decision-making processes in university, it is essential to acknowledge the principal position of STPs and to optimize their performance evaluation system. Second, universities should implement a distribution policy oriented toward enhancing the value of knowledge, increase the share of income from technology transfer for researchers, and fully stimulate their enthusiasm for innovation and entrepreneurship.

Due to data availability constraints, this study has notable limitations. The academic environment of universities was primarily measured through the subjective evaluations of STPs, which inevitably introduces the influence of personal values and emotions. Future research could attempt to construct an objective indicator system to assess the academic environment of universities. Expanding the research scope to compare differences in academic environments across universities of different tiers would also constitute a meaningful research topic.

## Figures and Tables

**Figure 1 behavsci-15-01475-f001:**
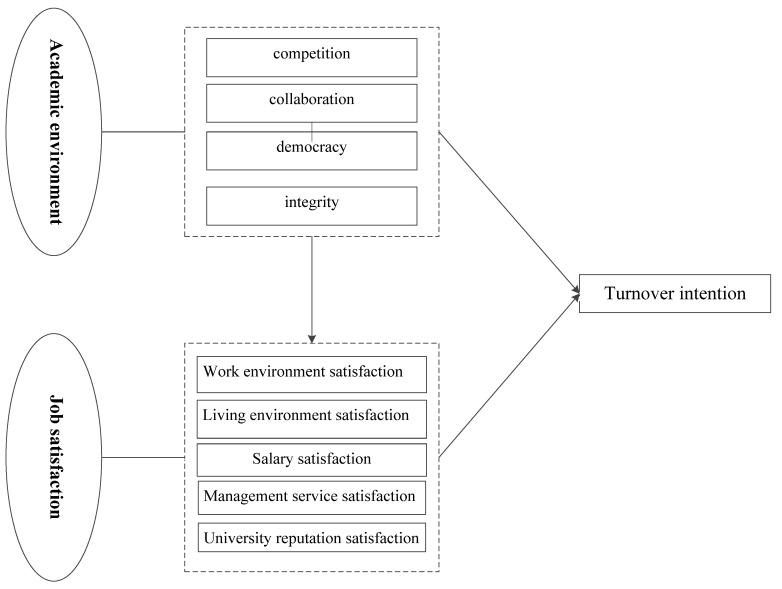
Research framework of the effects of academic environment and job satisfaction on turnover intention of high-skilled STP.

**Table 1 behavsci-15-01475-t001:** Analysis of the status quo of high-caliber scientific and technological talents’ flow intention, perception of academic ecology, and job satisfaction.

Variables		Overall (n = 482)	Eastern Region (n = 281)	Central Region (n = 122)	Western Region (n = 79)	*p* Value
Turnover intention		2.35 (0.04)	2.26 (0.88)	2.43 (0.78)	2.56 (0.78)	<0.01
Academic environment	Collaboration	3.33 (0.05)	3.28 (1.11)	3.38 (0.97)	3.41 (0.98)	0.11
	Competition	3.94 (0.04)	3.89 (0.94)	4.00 (0.76)	4.00 (0.93)	0.08
	Democracy	3.23 (0.05)	3.22 (1.11)	3.19 (1.07)	3.32 (0.94)	0.10
	Integrity	3.29 (0.05)	3.28 (1.09)	3.28 (1.08)	3.33 (0.96)	0.06
Job satisfaction	Salary	3.21 (0.04)	3.14 (0.89)	3.35 (0.81)	3.26 (0.73)	<0.05
	Work environment	3.42 (0.04)	3.51 (0.89)	3.37 (0.85)	3.17 (0.92)	<0.05
	Living environment	3.39 (0.04)	3.36 (0.86)	3.51 (0.68)	3.32 (0.76)	0.14
	University reputation	3.51 (0.04)	3.59 (0.93)	3.35 (0.83)	3.42 (0.83)	<0.05
	Management service	3.31 (0.05)	3.39 (1.03)	3.19 (0.93)	3.23 (1.06)	<0.05

Note: The *p* value is calculated from the results of analysis of variance.

**Table 2 behavsci-15-01475-t002:** OLS estimates for academic environment and job satisfaction on turnover intention.

	Variable	Model 1	Model 2	Model 3
Academic environment		−0.146 *** (−3.37)		−0.063 (−1.64)
Job satisfaction	Salary		−0.180 *** (−3.34)	−0.184 *** (−3.40)
	Work environment		−0.259 *** (−5.02)	−0.274 *** (−5.17)
	Living environment		−0.023 (−0.41)	−0.014 (−0.24)
	University reputation		−0.104 * (−2.13)	−0.094 (−1.92)
	Management service		−0.034 (−0.80)	−0.031 (−0.70)
Controlled variables	Gender (Male)	0.081 (0.69)	0.006 (0.06)	0.084 (0.82)
	Years of working	−0.000 (−0.08)	0.002 (0.61)	0.002 (0.47)
	Location (reference group: east region)			
	Central region	0.192 * (2.10)	0.151 (1.83)	0.163 * (1.97)
	Western region	0.356 ** (3.17)	0.215 * (2.22)	0.286 ** (2.89)
	Research platform (reference group: national level)			
	Ministerial and provincial level	−0.063 (−0.74)	−0.072 (−0.96)	−0.095 (−1.26)
	Other	0.195 (1.73)	0.159 (1.60)	0.132 (1.33)
Constant		2.744 *** (15.19)	4.478 *** (20.28)	4.478 *** (20.28)
R-squared		0.06	0.28	0.30
F		3.82 ***	15.90 ***	15.52 ***
N		482	482	482

Note: * *p* < 0.05, ** *p* < 0.01, *** *p* < 0.001.

**Table 3 behavsci-15-01475-t003:** Analysis of the mediating effect of job satisfaction in the effect of academic environment on the turnover intention of high-skilled STPs.

Academic Environment	Bootstrap		95% Confidence Interval	
	Coefficient	Standard Deviation	Lower Limit	Upper Limit
Collaboration				
Direct effect	−0.046	0.034	−0.112	0.021
Mediating effect of job satisfaction	−0.054	0.020	−0.096	−0.018
Competition				
Direct effect	−0.097	0.040	−0.176	−0.019
Mediating effect of job satisfaction	−0.045	0.022	−0.089	−0.022
Democracy				
Direct effect	−0.058	0.034	−0.125	0.008
Mediating effect of job satisfaction	−0.065	0.019	−0.105	−0.030
Integrity				
Direct effect	−0.046	0.033	−0.112	−0.120
Mediating effect of job satisfaction	−0.049	0.019	−0.088	−0.012

## Data Availability

The data underlying the findings of the study are available from the corresponding author upon request.

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
