# Peer review of "The Effect of Academic Environment on Turnover Intention of High-Skilled Scientific and Technology Professionals: Empirical Evidence from China’s First-Class Universities"

_behavsci, 2025, doi:10.3390/bs15111475_

Round 1

Reviewer 1 Report

Comments and Suggestions for Authors

Thank you for the opportunity to review the manuscript titled 'The Effect of Academic Environment on Turnover Intention of High-skilled Scientific and Technology Talent: Empirical Evidence from China’s First-class Universities.' This is an important topic, and the manuscript makes a valuable contribution to both the field and our understanding of factors influencing turnover intention among high-skilled talent. The research findings are insightful; however, there are several areas that would benefit from further improvement to strengthen the clarity, coherence, and overall impact of the study.

Below, I provide several recommendations and suggestions that, if addressed, could enhance the manuscript’s clarity, rigor, and contribution.

The abstract reads well. There are a few points here that I’ recommend authors to work on or revise:

  • What is the theoretical framework underpinning the research question and hypothesis? This needs to be added to the abstract
  • I’d recommend keeping the findings as high level, no need to present any statistics (e.g. line 3.94 mentioned in bracket needs to be removed. It’s important to elaborate the answer to the research question here.
  • Please add a high-level recommendation based on the findings of this research to the final line of the abstract.

Introduction:

  • Would be valuable, to define who scientific and technological professionals are? Are they laboratory managers, those working in research offices? Technicians? Faculty and department managers? Or generally non-academics working at the universities studied in this research project? Or it refers to both academics and non-academics working at universities.
  • At the end of introduction, please elaborate the contribution of this manuscript- what are the gaps this research is trying to fill? what are the main significance of this work

Literature Review:

    • I’d encourage the authors to elaborate how the literature review has been done what the scope of the review was, e.g. whether they have reviewed the literature on China or globally, etc.
    • It would be valuable to start with reviewing the environment of the Double First-class universities, followed by providing overview of the research so far on that have examined the turnover of high skilled professionals and, as these are the central concepts and focus of research questions. The discussion of the relationship between job satisfaction and the turnover intention of high-skilled STP is properly done, and it would strengthen the paper to develop the other sections in a similar way.
    • In the first sub-section of “The relationship between academic environment and the turnover intention of high-skilled STP” in the literature review, only one citation is provided, which is not sufficient to demonstrate what the broader literature says on the topic. Similarly on page 4 under the sub-title, The mediating role of job satisfaction, the review of the literature could be more thorough.
  • There is no mention of theories, what theory is this research and its hypotheses grounded in? The manuscript presents multiple hypotheses, yet these are not anchored in a clear theoretical framework. In quantitative research, hypotheses are typically derived from theory, which provides the rationale for testing specific relationships. If the intention is to conduct an exploratory study without a guiding theory, this should be stated explicitly and justified, as it shapes how the contribution is positioned within the literature. Clarifying this point would strengthen the manuscript’s coherence and scholarly value.

Methods:

  • Thank you for providing a definition of STP, it’s very helpful for understanding the context. I encourage the authors to present this information earlier in the manuscript so that readers do not need to speculate about its meaning.
  • Please elaborate how you ensured credibility and reliability of the collected data.
  • Authors need to clarify which platform was used to collect the data, as well as the statistical tools employed for analysis, such as SPSS.

Result analysis: I have shared a few general comments here for authors:

  • The section mixes descriptive statistics, ANOVA results, regression analyses, and mediation tests, which can make it difficult for readers to follow. Consider breaking it into clearly labeled sub-sections (e.g., Descriptive Statistics, Regional Differences, Regression Analysis, Mediation Analysis).
  • While the findings are described in detail, the section is largely descriptive. It would be strengthened by briefly interpreting the results in the context of prior literature or theoretical expectations. For example, why might eastern region talents have higher satisfaction with work environment but lower salary satisfaction? Linking findings to existing studies or conceptual reasoning helps readers understand implications.
  • Similarly, Comparative insights (e.g., differences between competition and collaboration scores) could also be tied back to literature on academic environment and turnover.

Conclusion and discussion:

  • It is common to separate the discussion from the conclusion. The discussion could be strengthened by integrating it with the presentation of the findings, drawing on the literature to show the extent to which the results align with or contrast existing studies.
  • Overall, the conclusion section is well written. In the last paragraph, it would be helpful to provide more elaboration on the implications for optimizing the academic environment, beyond suggesting improvements to the governance system. Based on the findings of this research, how would you recommend these improvements be implemented? Providing more practical implications and actionable recommendations would strengthen this part.
  • Additionally, please include any limitations of the study and suggest future research directions in the field.

Formatting:

  • Page 3, under literature review, the first line (The relationship between academic environment and the turnover intention of high-skilled STP) need to be italic for consistency with other section in literature review.
  • The formatting of the “Result analysis” section need to be consistent with other sub-titles.

Reviewer 2 Report

Comments and Suggestions for Authors

Dear authors,

I have read your manuscript very carefully. At the beginning of my review, I would like to state that this is a topic suitable for this scientific journal, as well as beneficial for both theory and practice. After analyzing the originality control protocol, I conclude that this is an original work, partial similarities with other works are absolutely negligible.

I have several comments and recommendations regarding the content of the manuscript. I believe that not all of its parts fully comply with the requirements of the instructions for authors published on the website of the scientific journal. The scope of 13 pages is appropriate, but based on my comments, I assume its expansion.
In addition to the fact that the chapters are not numbered, there is also chaos in the marking of individual pages. Page 10 of the manuscript is numbered 2 of 13 in the upper right corner, and the entire manuscript, or rather, its pages end with numbering 5 of 13.

However, I have constructive suggestions for the content of the manuscript, since not all of its parts meet the requirements of the instructions for authors published on the website of the scientific journal.

I recommend doing:

The abstract is like an initial introduction, which is not correct. It has to capture the reader's attention in 250 words. Therefore, it should contain a clear objective, the scientific research methods used, the results and conclusions (recommendations). In my opinion, it would be appropriate for the abstract to also mention the sources/data used, from which you drew knowledge and performed calculations. In addition, there are stylistic shortcomings in the abstract itself, as well as in the entire scientific study, which can be confusing for a layperson. What does "intention to leave" mean? to change the occupation of a university teacher to a manager? to an expert? or just to change the employer, i.e. the university? It may seem obvious to authors, but it is also necessary to take into account the legislation. It is necessary to clearly define terms such as work, employment, employee, employer and strictly adhere to these terms. In this regard, I must remind authors that even if they are investigating an issue in the field of personnel management, they must still remember that even a personnel manager can only operate within the limits of labor law.

The introduction should briefly place your scientific study in the issue and also emphasize its importance. However, I see the problem in that the introduction is very general and too long, there is no mention of the systematics of your manuscript. The introduction should also contain the established hypotheses, but these are established in other parts, which I consider sufficient.

On page 3 of 13, the chapter Literature Review and Research Hypothesis begins, although not marked with a serial number. It would be appropriate to supplement it with a deeper analysis of the researched issue, including citing key publications supplemented with other current and especially non-Chinese sources. Based on this opinion, I recommend supplementing the theoretical basis and therefore the number of sources with several current open access works in the field of labor law flexibility included in the world scientific databases WoS and Scopus, such as:

Peráček, T. (2021). Flexibility of creating and changing employment in the options of the Slovak Labor Code. Problems and Perspectives in Management, 19 (3), pp. 373-382. doi:10.21511/ppm.19(3).2021.30

Pessina, S. 2025. The Link Between Environmental Rights and the Rights of Nature:The Virtues of a Complexity-Based Approach. Juridical Tribune Review of Comparative and International Law, 15(2), pp. 406–422, doi: 10.62768/TBJ/2025/15/2/09

In the methodology, you only list some of the scientific research methods used, which are probably the most important for you. These are analysis and comparative analysis. You do not list other scientific research methods. This is incorrect. However, you forgot to devote space and attention to the basic scientific research methods used by you, such as synthesis, deduction, induction, including their brief description and justification for their use for each chapter of the manuscript, as stated in the work Peráček 2021.

On page 10, the last chapter is marked as Conclusion and Discussion, which is not legal. The correct one should be Discussion and Conclusion. In addition, you are missing the discussion itself, as required by the instructions for authors, and the established hypotheses should also be confirmed/rejected here. At the end of this chapter there should be at least a brief mention of the limits of this research as well as the possibilities and directions of your potential future research.

Finally, I would just like to remind you of the need to unify the designation of your scientific study. In the abstract you use the term “study” in other places of the manuscript “paper”. Given the nature and scope of the manuscript, I recommend using the term “study”.

Reviewer 3 Report

Comments and Suggestions for Authors

The paper covers an interesting topic and is clearly structured, but some improvements are needed:

  • I consider that a more concise formulation should be found for the title.
  • In the abstract, the research findings should be presented in a general manner, without details.
  • The literature review should be expanded by identifying and analyzing works that address similar topics. Additionally, the variables of the research model should be more grounded.
  • The discussion section should be much more consistent. The authors' findings should be compared to those of other works addressing similar topics.
  • The theoretical and practical implications of the study and its limits should be presented in more detail.

Round 2

Reviewer 1 Report

Comments and Suggestions for Authors

Thank you for your thoughtful response to my earlier comments. I can see that the revisions you have made have strengthened the manuscript, and I appreciate the effort you have invested in addressing the feedback.

That said, I remain concerned about the treatment of theory. While you note that the study draws on academic ecology theory and job satisfaction theory, the manuscript does not yet provide an explanation of these theories, their key concepts, or how they inform the hypotheses. Naming the theories alone is not sufficient- readers need to see the theoretical grounding and logical link between the concepts and the hypotheses. To ensure the hypotheses are clearly grounded and the contribution of the study is well articulated, I recommend that you expand this section by briefly introducing each theory and explicitly linking their constructs to your hypotheses. Doing so will enhance the manuscript’s coherence and scholarly impact.

Thank you for the valuable contribution your manuscript makes to the field.

Author Response

Dear reviewer,

Thank you for your thoughtful and careful re-review of the revised manuscript. We fully agree with your suggestion regarding the theory. Below, we provide a detailed explanation of the changes made.

Comment 1: That said, I remain concerned about the treatment of theory. While you note that the study draws on academic ecology theory and job satisfaction theory, the manuscript does not yet provide an explanation of these theories, their key concepts, or how they inform the hypotheses. Naming the theories alone is not sufficient- readers need to see the theoretical grounding and logical link between the concepts and the hypotheses. To ensure the hypotheses are clearly grounded and the contribution of the study is well articulated, I recommend that you expand this section by briefly introducing each theory and explicitly linking their constructs to your hypotheses. Doing so will enhance the manuscript’s coherence and scholarly impact.

Response 1: We completely agree with your point that the introduction of a theoretical foundation is crucial for formulating research hypotheses in empirical studies. Unfortunately, we were unable to find a theoretical framework that could be directly applied to the development of our research hypotheses. Therefore, we have made every effort to strengthen the explanation of the theoretical foundation and its application. First, in the third paragraph of the introduction, we have introduced the academic ecosystem and its components, analyzing the impact of the academic environment on research outcomes. When formulating Hypothesis 1, we further examined the relationship between the academic ecosystem theory and the research hypotheses. Second, in the formulation of Hypothesis 2, we enhanced the elaboration of the job satisfaction theory. Third, in analyzing the mediating role of job satisfaction, we categorized the academic environment as external supportive conditions, and defined job satisfaction as an individual’s internal evaluation of factors such as the work environment, living environment, salary level, and management services. External conditions exert their influence through these internal individual factors.

Best regards

Authors

Reviewer 2 Report

Comments and Suggestions for Authors

Dear Authors,

Based on your review of the changes you have made, I agree to publish your manuscript.

Author Response

Dear reviewer,

We are deeply grateful for your thoughtful review and the invaluable feedback you provided regarding our manuscript titled “The Effect of Academic Environment on Turnover Intention of High-skilled Scientific and Technology Talent: Empirical Evidence from China’s First-class Universities”. We hope our research contributes to advancing progress in this field.

Best regards

Authors

Reviewer 3 Report

Comments and Suggestions for Authors

In my view, the revisions introduced in the current version of the paper are rather superficial. Consequently, my initial overall recommendation remains unchanged.

Author Response

Dear reviewer,

Thank you for your thoughtful and careful re-review of the revised manuscript. In response to your initial revision suggestions, we have carefully revised and improved the manuscript accordingly.

Comment 1: I consider that a more concise formulation should be found for the title.

Response 1: We have revised the keywords in the title to ensure consistency with the content of the manuscript.

Comment 2: In the abstract, the research findings should be presented in a general manner, without details.

Response 2: We have removed the mention of specific numerical data from the abstract and revised it accordingly.

Comment 3: The literature review should be expanded by identifying and analyzing works that address similar topics. Additionally, the variables of the research model should be more grounded.

Response 3: In the literature review and hypothesis  sections, we have enhanced the elaboration of the academic ecosystem theory and the job satisfaction theory.

Comment 4: The discussion section should be much more consistent. The authors' findings should be compared to those of other works addressing similar topics.

Response 4: We fully agree with your suggestion. Due to the limited existing research on the relationship between the academic environment and STP’s turnover intentions, the comparison with prior studies in that section is relatively limited. In our discussion of how job satisfaction effect turnover intention, we have provided a more comprehensive comparison with existing literature.

Comment 5: The theoretical and practical implications of the study and its limits should be presented in more detail.

Response 5: In the conclusion section, we have strengthened the discussion of the practical implications and limitations of the study.

Comment 6: The English could be improved to more clearly express the research.

Response 6: We have reviewed and improved the English language of the manuscript, and also utilized the author services provided by the journal to further enhance its clarity and professionalism.

Best regards

Authors